# The co-management of HIV and chronic non-communicable diseases in the Dominican Republic: A qualitative study

**Deshira D. Wallace** [1]◉*, **Amarilis Then-Paulino**[2‡], **Gipsy Jiménez Paulino**[3‡], **Flabia Tejada Castro**[4‡], **Stephanie Daniela Castro**[5‡], **Kartika Palar**[6‡], **Kathryn P. Derose**[7,8◉]

1 Department of Health Behavior, Gillings School of Global Public Health, University of North Carolina at Chapel Hill, Chapel Hill, North Carolina, United States of America, 2 Instituto de Investigación en Salud de la Facultad de Ciencias de la Salud, Facultad de Ciencias de la Salud, Universidad Autónoma de Santo Domingo, Alma Máter, Ciudad Universitaria, Santo Domingo, Dominican Republic, 3 Viceministerio de Garantía de la Calidad, Ministerio de Salud Pública, Santo Domingo, Dominican Republic, 4 Independent Consultant, Santo Domingo, Dominican Republic, 5 Center for Diagnosis, Advanced Medicine, and Telemedicine (CEDIMAT), Santo Domingo, Dominican Republic, 6 Department of Medicine, Division of HIV, Infectious Disease, and Global Medicine, University of California, San Francisco, San Francisco, California, United States of America, 7 Department of Health Promotion & Policy, University of Massachusetts Amherst, Amherst, Massachusetts, United States of America, 8 Department of Behavioral and Policy Sciences, RAND Corporation, Santa Monica, California, United States of America

◉ These authors contributed equally to this work.
‡ ATP, GJP, FTC, SDC and KP also contributed equally to this work.
* ddwallac@email.unc.edu

**Data Availability Statement:** The data upon which this manuscript is based are in-depth interview transcripts from a small sample of marginalized

## Abstract

People living with HIV and a non-communicable disease (NCD) experience multi-level barriers when co-managing multiple conditions. We explored the factors affecting living with multiple chronic conditions in the Dominican Republic. We conducted 21 in-depth interviews from October 2019-February 2020 with Dominican adults who participated in a food security intervention and managed HIV and at least one chronic NCD. Using thematic analysis, we explored participant lived experiences co-managing multiple chronic conditions. All participants (mean age = 45.5 years) were linked to HIV care, but only three were linked to NCD-specific care. Individual-level barriers to managing NCDs included limited education and limited self-efficacy for self-management. Interpersonally, barriers included limited rapport building with an NCD-specific specialist. Structural barriers to managing NCDs were no health insurance, poor referral systems, and limited financial assistance. Health system adaptation requires equitably considering the needs of individuals managing multiple chronic conditions. Key factors to address include patient-provider relationships, improved referral systems, accessibility and availability of specialists, and financial assistance.

## Introduction

The Caribbean region is undergoing a health transition in which traditionally infectious diseases are still occurring alongside with a rise of chronic non-communicable diseases (NCDs). As of 2019, the Caribbean recorded the second highest HIV prevalence globally, after sub-

and stigmatized people with HIV and contain highly sensitive data that are identifiable via inference. Thus, sharing them would violate the promise of confidentiality made to participants during informed consent. Because we did not state in the consent form that the data could be made available to outside researchers, the RAND Human Subjects Protection Committee (HSPC) has informed us that we cannot make the data available outside the study team. Queries regarding data access can be addressed by the RAND HSPC by contacting Rebecca Collins, HSPC Chair (collins@rand.org).

**Funding:** Research reported in this publication was supported by the National Institute of Mental Health of the National Institutes of Health under Award Number R34MH110325 (KPD). DW was supported by NICHD of the National Institutes of Health under award number P2C HD050924. KP contributions were supported by a grant from the National Institutes of Health, University of California, San Francisco-Gladstone Institute of Virology & Immunology Center for AIDS Research, P30AI027763 (NIAID). The article contents are solely the responsibility of the authors and do not represent the official views of the National Institutes of Health. The funders had no role in study design, data collection and analysis, decision to publish, or preparation of the manuscript.

**Competing interests:** The authors have declared that no competing interests exist.

Saharan Africa, with approximately 1.1% of people in the Caribbean are living with HIV [1, 2]. Within the Caribbean, the Dominican Republic has a similarly high HIV prevalence at approximately 0.9% of the population [3, 4]. In fact, HIV/AIDS-related complications are the fifth cause of death in the country [5]. Recent evidence shows that 82% of people living with HIV (PLHIV) in the Dominican Republic know their status and about 51% of PLHIV there report being on antiretroviral therapy (ART) [3]. Although HIV is an infectious disease, advances in treatment have shifted HIV to function more as a chronic condition, requiring similar behaviors for its management such as medication adherence, lifestyle adjustments, consistent and long-term provider interaction, and adequate interpersonal support [6].

As related to chronic NCDs, diabetes and cardiovascular diseases are among the leading causes of death and morbidity globally [7]. Chronic NCDs include a range of conditions such as cardiovascular diseases, type 2 diabetes mellitus (T2D), cancer, pulmonary diseases, neurological conditions, and mental health conditions. Trends in the global rise of chronic NCDs are similar in the Dominican Republic [8], such as ischemic heart disease, intracerebral hemorrhage, and ischemic stroke reported as three of the four top causes of death in the country [5]. Overall, by 2019, chronic NCDs accounted for 72% of deaths in the country, a 40% increase since 2010 [9].

With HIV being managed as a chronic condition, and the continued rise of chronic NCDs, there is acknowledgement that these conditions do not happen in silos. Thus, a shift in models of care that account for the management of multiple chronic conditions is necessary. However, as individuals with multiple chronic conditions represent an increasing proportion of the adult population worldwide, studies have found that multilevel barriers affect the effectiveness of care for these individuals [10, 11]. At the individual level, those with multiple chronic conditions require greater healthcare utilization; therefore, incurring greater healthcare costs than individuals with one or zero chronic conditions [12]. The costs associated with high utilization can stem from emergency room use, inpatient hospitalizations, outpatient consultations and care, and medication use [10]. The cost of managing HIV and chronic NCDs disproportionately affect individuals who are socially and economically marginalized. Specifically, disease self-management (e.g., dietary changes, medication adherence) for persons experiencing intersecting marginalized social and economic conditions, such as food insecurity, require using a significantly higher proportion of their available income compared to individuals who are economically secure and do not experience various social forms of marginalization and stigma [13, 14]. Interpersonally, individuals with multiple chronic conditions require sufficient social support from family, friends, and their healthcare team (i.e., patient-provider relationships) to manage their care [15, 16]. At the structural level, clear coordination between primary care providers and specialists, effective coordination among specialists to ensure suggested treatments are aligned, healthcare accessibility (both distance and financial), and healthcare availability are important for the management of each condition [10]. These barriers are further complicated in low-and-middle income countries (LMICs) where the health systems are focused on addressing burdens of communicable diseases such as vector-borne conditions and HIV, as well as maternal, child and neonatal health [17]. Studies from Ethiopia and Brazil have found an increasing number of people living with HIV (PLHIV) are not meeting glycemic control targets [18–20]. A study among 340 PLHIV in Brazil found a significant number of respondents were dealing with obesity, high cholesterol, and a significantly higher likelihood of metabolic syndrome [21]. A recent meta-analysis estimated that the regional prevalence of hypertension among PLHIV in Latin America and the Caribbean at 22% (17.8–26.5) indicating the increasingly overlapping health needs of PLHIV [22, 23]. Therefore, accounting for individual, interpersonal, and structural facilitators and barriers to disease co-management is needed when evaluating living with multiple chronic conditions.

There are rich data on the experiences of managing HIV across several sub-populations in the Dominican Republic (e.g., urban versus rural, by occupation) [24–27], and there are increasing numbers of studies on T2D and cardiovascular diseases in the Dominican Republic [28–30]. However, to our knowledge, there are little data on the co-management of these conditions with HIV, particularly in a country with substantial investment in HIV care while balancing the rising burden of NCDs. Furthermore, food insecurity has been shown to affect management of both HIV and chronic NCDs such as T2D. In fact, food insecurity adversely affects HIV outcomes and NCDs through a variety of behavioral, nutritional, and psychosocial pathways [31, 32]. Understanding the realities of adults in structurally vulnerable food insecurity states can improve our understanding of ways to adapt systems to improve disease co-management for broader segments of the population. Therefore, we were interested in exploring how adults experiencing food insecurity, and thus living in more resource-constrained environments, are managing multiple chronic conditions in the Dominican Republic.

We apply the socioecological model to explore qualitative interviews with Dominican adults experiencing moderate-to-severe food insecurity and managing multiple chronic conditions. Themes focus on individual level, interpersonal level, and structural level factors affecting participant health and wellbeing. The contribution of this work is that it provides formative information on the considerations of public health approaches to address multiple condition management in a Dominican or similar contexts.

## Materials and methods

### Study

This qualitative study is part of an endline evaluation of a 12-month pilot intervention led by researchers at the RAND Corporation, a non-profit research organization, and the School of Health Sciences at the Universidad Autónoma de Santo Domingo (UASD). The intervention (i.e., ProMeSA) integrated urban gardens and peer nutritional counseling for adults with HIV experiencing moderate or severe food insecurity and treatment adherence difficulties in urban centers in the Dominican Republic [33]. The intervention comprised gardening training, home-based agricultural follow-up, peer nutritional counseling, and a group garden-based cooking workshop. The participants underwent two follow-up periods to assess potential changes HIV clinical outcomes (viral load, ART adherence, and HIV care retention) and other HIV-related outcomes (internalized and experienced stigma, social support).

The endline qualitative study involved in-depth interviews with patients participating in the pilot program at the intervention clinic (another clinic served as a usual care control). The broader goal of the interviews involved exploring health care experiences, facilitators and barriers to managing HIV, facilitators and barriers to managing a chronic NCD, and participants' perceptions regarding the feasibility, acceptability, and impacts of the intervention. This article focuses on the facilitators and barriers of co-managing HIV and a chronic NCD using the endline qualitative data.

### Setting and participants

The intervention study, ProMeSA, was conducted at two HIV clinics located in the central and northwestern part of the Dominican Republic [33]. The clinics are approximately a 2-hour drive from each other to avoid cross-contamination between participants. Both clinics were selected as they were of similar sizes, 500–800 patients on ART, and offered similar standards of care. Participants in the intervention study were 18 years and older, registered patients at the HIV clinic sites, resided in an urban or peri-urban area, had moderate or severe food insecurity, and had evidence of adherence difficulties. For the qualitative endline sub-study, we

used purposive sampling [34] to select intervention participants who also had a chronic NCD (i.e., hypertension, diabetes, prediabetes, high cholesterol, obesity) according to previously collected data. We purposively recruited approximately equal numbers of men and women in-person. Furthermore, participants were recruited until we reached thematic saturation. Institutional Review Boards and ethics committees from RAND, UASD, and the Dominican Ministry of Public Health approved the study protocols. All intervention study participants provided written consent; for the additional qualitative endline interview, participants provided verbal consent to account for literacy and accessibility.

## Data collection

Three Dominican medical school graduates (one as intervention study coordinator [GJP] and two as research assistants [FTC, SDC]) were trained in qualitative data collection and analysis by KD and DW prior to and throughout the study period. This involved virtual and in-person trainings in Santo Domingo on how to conduct semi-structured interviews, manage data, transcribe audio recordings, and code transcripts. We used semi-structured, in-depth interviews to explore our themes of interest for this study: health experiences living with HIV, health experiences living with a chronic NCD, and health services experiences with both conditions. Using a socioecological model, we framed questions to understand individual, interpersonal, organizational, and political barriers and facilitators to managing multiple chronic conditions. Example questions included: how are you managing your HIV? How are you managing your chronic condition (T2D, high blood pressure)? Which of your health conditions worries you the most?

Qualitative interviews were completed between October 2019 –February 2020 at the intervention clinic in the Dominican Republic. All interviews were conducted in Spanish, lasted between 45–90 minutes, and were audio recorded with participant consent. Immediately after each interview, interviewers wrote field notes about the interview setting, key themes of the interview, and observations for the following interviews. Audio recordings were transcribed verbatim by the three trained interviewers/research assistants (GJC, FTC, DC) and verified by the first author.

## Data analysis

We used a thematic analysis approach to explore participants' lived experiences co-managing multiple health conditions through the examination of deductive and inductive codes [35]. To do that, we first uploaded transcripts to Dedoose, a web-based text management and analysis software [36]. We then developed a codebook, which was initially developed using questions from the interview guide and field note observations. The codebook was iteratively updated based on the coding of initial transcripts. Four trained coders (DW, GJP, FTC, DC) read and coded the transcripts independently and the first author conducted a secondary coding review for dependability. Transcripts coded by the first author were reviewed by another member of the coding team. When reading the transcripts, coders wrote memos, or notes, about their observations of the data and linked what they read with other transcripts. All coding and analyses were completed in Spanish. Any discrepancies or questions about coding were resolved via coder consensus.

Some key codes we used to analyze participant co-management experiences were: how participants were diagnosed with HIV and the chronic NCD, emotional reactions to their diagnoses, their understanding/knowledge of either condition, how they managed their conditions, social support with management, and their experiences with the healthcare system for both conditions. After coding, all codes were placed in a matrix organized by participant to facilitate

constant comparison of experiences among participants and within codes. We constructed summaries by code to examine similarities and differences in participant experiences, and then combined codes to construct overarching themes. Themes are organized using a socioe-cological framework approach, with consideration for individual, interpersonal, and structural facilitators and barriers to disease co-management in the Dominican Republic. Illustrative quotations that highlighted themes were selected and translated into English by the bilingual authors.

## Results

A total of 21 participants were recruited, of which 12 were men and 9 were women. The mean age of participants was 45.5 years old. Eighteen participants self-identified as being of Domini-can descent and three self-identified as being of Haitian descent. While all participants were diagnosed with HIV, participants had a range of chronic NCDs. These conditions included T2D, hypertension, high cholesterol, and cardiovascular and circulatory diseases (see Table 1). Note that individuals could be listed for multiple chronic NCDs.

The results are organized by providing a narrative of HIV management, narrative of chronic NCD management, and how these conditions compare in terms of their ability to co-manage them at the individual, interpersonal, and structural levels. Since only individuals who had experienced moderate or severe food insecurity were enrolled in the intervention study, all narratives are filtered through those experiences.

### Living with HIV

**Diagnosis.** Participant narratives all began at diagnosis. All but three participants were diagnosed at a clinic or hospital, and the remaining three had a workplace screening that trig-gered a clinic referral. The reasons for initial HIV testing varied by gender with women describing getting tested for two reasons. The first was getting tested because their partner or

**Table 1. Participant demographics (n = 21).**

| Variable | N | % | Mean |
|---|---|---|---|
| Age (years) | | | 45.5 |
| Female | 9 | 43.0 | |
| Ancestry | | | |
| Dominican | 18 | 86.0 | |
| Haitian | 3 | 14.0 | |
| Active Insurance Coverage | 14 | 67.0 | |
| Non-Communicable Chronic Conditions | | | |
| Prediabetes | 5 | 24.0 | |
| Diabetes | 5 | 24.0 | |
| Overweight/Obesity | 12 | 57.0 | |
| Hypertension | 7 | 33.0 | |
| High Cholesterol | 3 | 14.0 | |
| Cardiovascular disease | 1 | 4.8 | |
| Thrombosis | 1 | 4.8 | |
| Number of Chronic Conditions | | | |
| 1 | 10 | 47.6 | |
| 2 | 7 | 33.3 | |
| 3 | 4 | 19.0 | |

spouse received a positive test, had HIV-related complications, or passed away due to untreated HIV complications. The second reason was pregnancy as two women tested positive after their second trimester tests. In contrast, men often got tested because they fell sick and went to a clinic where they learned their status and a few men went in for regular HIV/STI testing and received their diagnosis then.

Relatedly, receiving an HIV diagnosis caused a range of emotions, from indifference to sadness. Overwhelmingly the reactions were negative. The most common was sadness about having to live with HIV (n = 8). Participants also mentioned experiencing depressive symptoms, which stemmed from their belief that HIV was a "death sentence" (n = 5). The reactions differed by gender, such that women all noted that they were sad, devastated, or depressed by the news. Whereas some men did mention shock, denial, and depression, others expressed feeling as if were in "limbo" about their ability to work and maintain their livelihoods. Specifically, they worried if they would continue to have the physical strength to maintain their job.

**Management.**    At the time of the interview, all participants stated that they were engaged in medication taking and treatment. Although several noted not immediately taking medications because they felt fine when diagnosed or because they initially had adverse reactions to the medications. Participants mentioned that they could live a normal life with HIV as long as they took their medications, went to the doctor, practiced safe sexual practices, and took care of their nutritional needs. The automation of self-management led participants to not overly think or worry about their HIV (*no dar mente*), which was important because several participants linked the rumination over their HIV status with the development of depressive symptoms. A few participants explicitly mentioned that having access to government social services (e.g., subsidized medication, food supplement program) also enabled them to manage their condition. There were two primary challenges participants faced when managing their HIV. The first was stigma, which men spoke more often about, specifically judgement from others who held inaccurate beliefs about HIV. Persons holding stigmatizing beliefs could be family, friends, neighbors, and colleagues, which complicated who they could receive social support from. Second, both men and women spoke equally about the related difficulty of maintaining partners as participants experienced rejections and judgments from potential partners once they disclosed their HIV status.

## Living with a non-communicable chronic condition

**Diagnosis.**    Most participants were diagnosed with chronic NCDs at a clinic or hospital. Several participants (n = 8) were diagnosed when they had to go to a health care facility because they felt ill, such as feeling dizzy, excessively tired, "suffocated", or experienced chest pains. Several participants (n = 5) noted that they were diagnosed with a chronic condition during a HIV treatment follow-up at a clinic. For these participants, their only healthcare provider was their HIV provider who may have offered dietary and physical activity guidance. Participant responses to their diagnosis were primarily negative. Sadness and fear were the most common reactions as participants cited knowing family or friends who had passed away from a related condition (e.g., T2D, hypertension). Relatedly, inevitability was another reaction particularly if there were several family members who also lived with the condition.

**Management.**    At the time of the interview, only three participants were able to follow-up on their respective diagnosis with a chronic condition-specific provider. Implications of those who could not follow-up with providers meant that participants were delayed in establishing a treatment plan to address the specific condition. As one participant, a 53-year-old man with T2D and hypertension, explained his difficulties with getting a formal diagnosis affected his ability to begin treatment:

Every time I come to the [HIV] clinic my blood sugar is high. But I have yet to get a diagnosis so I can't see a specialist to see if I need medication or not, because at this time I am not taking anything.

For those who were diagnosed and linked to care, they often spoke of needing support to manage their medications and key behaviors, primarily diet and physical activity. Five facilitators that participants described as helping them self-manage their chronic conditions were: reduced medication costs; dietary education; improved social support from friends and family; referrals or access to specialists; and maintaining a "calm" demeanor. Of the five facilitators, the notion of maintaining a calm demeanor as a practice came from participants with either hypertension or T2D. For participants with hypertension, emotional regulation was tied to the belief that blood pressure is sensitive to emotions. For those with T2D, the explanation was that T2D caused individuals to be "hot blooded", a stigmatizing belief; therefore, they needed to consciously work on regulating their emotions for their health.

## Living with the double burden of conditions

Participants explained that HIV had its challenges with management; however, there was a consensus among all participants that managing a chronic NCD was more difficult for them. A few participants specifically mentioned that they had trouble getting their chronic NCDs "under control". For example, a couple participants with hypertension described how they had to seek care at a clinic or emergency room rather recently due to blood pressure dysregulation, thus found a condition like hypertension to be much more unpredictable and difficult to manage than HIV. The perception of several participants when comparing HIV to another condition was that more people were dying from conditions such as T2D complications than HIV, making chronic NCDs more dangerous to live with. As mentioned by a 44-year-old man living with high cholesterol: "At the moment, I would say that HIV is easier to manage because I have medical care [for it] here at the hospital."

Even without the long-term view, there was a perception that T2D was worse than HIV because it would inevitably "spoil" your body from inside out, as described by a 41-year-old woman living with T2D:

With HIV you can live for some time when you take your medication, but with "sugar" it spoils you from the inside.

Conversely, participants believed that they could "live a normal life" with HIV if they took their ARTs consistently. There was some discussion of eating healthier and taking care of themselves in general, but all participants emphasized medication adherence as the primary form of HIV management. When participants described their experiences with certain chronic conditions, such as T2D and hypertension, medication use was also noted as an important component of self-management; however, medication use was not the only concern. The primary concern was that participants felt burdened by having to change their lifestyles. As one 52-year-old woman managing obesity described of T2D, "You cannot eat anything. You cannot eat much rice, and you cannot eat spaghetti." These restrictions were particularly difficult as foods like rice and spaghetti were more feasible to attain for participants as they were experiencing food insecurity. However, these were the types of food that were not recommended for them to consume as they managed their NCD. This was further expounded upon by a 56-year-old woman managing hypertension and T2D when explaining which condition was worse for her:

[Diabetes] because with HIV I take my pill and I feel fine. But with diabetes sometimes when one eats something their blood sugar immediately goes up. I must stay vigilant that I can't eat this or can't eat that. That's why it's more worrisome.

There were concerns about T2D-related complications such as vision loss and amputations. Similarly, hypertension caused fear because people could "die instantly" from one moment to the next. Overall, there was a feeling of not having as much control over these conditions as compared to HIV, where all participants felt relatively secure that HIV-related complications were no longer a threat to their well-being.

Participants provided examples of how their experiences co-managing these conditions varied at the individual, interpersonal, and structural level. A summary of these experiences is provided in Table 2 and in further detail in subsequent sections.

### Individual level: Self-management

As participants spoke of their self-management of multiple chronic conditions, there were distinct reasons that self-management was either straightforward or challenging. HIV self-management was most often described as straightforward primarily for two reasons. The first was

**Table 2. Multi-level participant experiences with co-managing conditions.**

| | HIV | Chronic Conditions (Non-communicable) |
|---|---|---|
| **Self-management** | Generally, participants described that it was easier to manage HIV due to:<br>• Better knowledge ("orientation"), practice, self-efficacy, and consistent access to medications to self-manage their HIV<br>• Primary behavior for HIV self-management was consistent medication taking | Consensus was that there were more difficulties to self-manage their chronic conditions for several reasons:<br>• The unpredictability of the daily changes of their conditions, such as daily blood pressure spiking or blood glucose increasing or decreasing<br>• Need for more "orientation" or knowledge on how to best self-manage<br>• Medication taking was inconsistent due to cost, ability to get (re)fills<br>• Several chronic conditions are sensitive to behaviors other than medication, such as diet, which was difficult to always do consistently |
| **Patient-provider** | • All participants had built rapport with their HIV providers and mentioned having good patient-provider interactions<br>  • Providers described as friendly, available (can answer questions after hours), and supportive<br>• Most participants named their HIV providers as their primary care doctor<br>• HIV providers did some basic follow-up that was relevant to specific conditions, particularly hypertension; however, it was not comprehensive care | • Most participants did not have a regular provider to address their chronic condition<br>• Rapport building with providers was limited due to inconsistency in attending different places |
| **Access to Care** | • Referral system once diagnosed with HIV was clear, therefore linkage to care was consistent<br>  • Participants felt that it was easier to get linked to care for HIV<br>• Participants attributed consistent HIV care with low or no cost for health appointments | • Several participants were diagnosed with T2D or had elevated blood glucose (i.e., prediabetes) but were not referred to a specialist<br>• Several participants used emergency room services to address any health emergencies (hypertension, T2D)<br>• Specialist care can be cost-prohibitive making it hard to establish care, build rapport, and have consistent follow-up to evaluate changes to the condition<br>• A few participants did not know where to seek care to address their conditions<br>• There were a range of care options participants used for chronic disease management<br>  • Government clinics (*policlinica*) when needed<br>  • Specialists when funds were available for clinic fees or for transportation (many specialists were in larger cities)<br>  • HIV providers if they had any general questions |

that participants believed that they had sufficient knowledge or "orientation" about what HIV was and what they needed to do to manage the condition, and they had enough time to practice the skills they needed to self-manage, which was related to the improved self-efficacy. Second, all participants mentioned that the consistent access to necessary medications facilitated their ability to adhere to their treatment plan.

On the other end, participants found self-management of their chronic NCDs to be more difficult. Unlike the amount of orientation they received for HIV, participants felt that they received limited education on how to best self-manage their specific chronic condition. The lack of education on NCD self-management was also tied to how participants experienced their condition as being "sensitive" and "unpredictable" from one day to the next. This is exemplified by a 48-year-old woman with T2D and hypertension.

> I manage HIV more easily than diabetes because with HIV I take my medication just fine. But with diabetes, even when I take the medication, sometimes based on what I eat, my blood sugar goes up [. . .] Also with diabetes, if you cut yourself the cut can start to rot–it's the worst disease. They say that HIV is bad, but the worst are cancer and diabetes.

As described above, hypertension and T2D were cited as being difficult as they were both very sensitive to dietary habits, and were prone to unpredictable extremes (e.g., hyperglycemia, uncontrolled high blood pressure) that would incapacitate participants or require them to seek emergency care. For NCDs, food was often cited as part of the healthcare regimen, equivalent to medication itself. Therefore, food insecurity complicated the ability to purchase the recommended foods and maintain a consistent diet.

## Interpersonal level: Patient-provider interactions

Interpersonal relationships were an important theme from the interviews. All participants described that they had "good" interactions with their HIV providers. Specifically, HIV providers were described as friendly, responsive at all times of the day, and supportive, as explained by a 34-year-old man managing obesity when speaking about his providers:

> [The doctors] are always looking after you, and they have what I would say–rather it makes me feel good. Because the [doctors] support us, they are always with us.

These positive-leaning relationships with their providers were noteworthy because most participants named their HIV providers as their primary care provider (*doctor de cabecera*). Participants described the range of services they received from their HIV providers, for example, they received standard HIV-related care, HIV providers evaluated their HIV medication adherence, and followed up on their progress living with HIV.

Only two participants stated that they had more than one medical provider. In addition to their HIV providers, both participants had access to a cardiologist to help them manage their hypertension. One participant explained that they met with their cardiologist on a consistent basis, whereas the second participant met with theirs on an inconsistent basis due to financial constraints related to paying the consultation fee. Of note, participants explained that neither provider discussed the co-management of both of their conditions.

## Structural level: Access to care

Both individual self-management and interpersonal interactions with providers are dependent on facilitators and barriers at the structural level. Structurally, a third of the study participants

did not have health insurance. Two participants lost coverage between the overall study's 6-month and 12-month follow-ups, however, their narratives regarding care seeking did not seem to change based on their change in health insurance status. Of the five participants who were without health insurance throughout the 12-month study period, three were the participants of Haitian descent. Of the 14 participants with health insurance, only three participants had a medical provider for each chronic condition. Overwhelmingly, participants with health insurance noted not being linked to care for their chronic NCD due to accessibility barriers. Thus, the narratives of participants with and without health insurance regarding their access to HIV and NCD care were similar.

The narratives on access to health care were distinct for HIV as compared to chronic NCDs. According to participants, the HIV system was clearer or "easier." Specifically, once participants were diagnosed with HIV, they all described getting linked to the appropriate hospital/clinic and healthcare team. Once linked to the appropriate HIV specialist, participants stated that they also were able to receive low or no cost consultations and medications. Participants attributed their consistent HIV self-management to the financial support they received that facilitated their health care access.

For their chronic NCD condition, the narratives were varied due to inconsistent referrals and poor or no financial assistance. This comparison was detailed by a 49-year-old man managing hypertension and cholesterol.

> HIV is easier because I just need to have transportation money to come here and do not need to pay for anything else. But for blood pressure, for cardio, I must always bring money because it's a private clinic.

First, participants did not present narratives detailing a consistent referral system after diagnosis. Several participants who were diagnosed with T2D or had elevated blood sugar (i.e., pre-diabetes) were not referred to specialists upon diagnosis. Further, a few participants with hypertension spoke about not having access to a specialist for their care.

When asked if participants knew where to seek care for their conditions the responses were either they did not know, the emergency room, or they spoke with their HIV providers. Emergency rooms were often stated as the primary site to address health needs as they arose and could be their primary source of information for managing their condition. This was especially the case for participants who described living too far away from specialists or never receiving a medical referral. This balance was explained by 53-year-old man living with hypertension and T2D:

> They send us to Santiago [a major city]. There is not a clinic here that you could say, let me go there because it is for diabetes or something. I think that for emergencies if you go to the emergency room and a doctor sees you and finds out that you have high blood pressure or some other problem, then you will get referred to see a specialist.

When it came to asking their HIV providers for help or follow-up on non-HIV conditions, the most common example was that HIV providers checked their blood pressure thus participants deemed these vital sign checks as part of their care for hypertension. As for financial resources to seek care, participants mentioned that they could only go to specialists when they could save enough money for the transportation to the appointment, the consult, and any prescribed medications. Relatedly, participants spoke about how job insecurity and limited financial resources made it harder to seek specialized care, maintain a recommended dietary regimen, and/or purchase necessary medications. The issue of the cost of medication was highlighted by a 48-year-old woman living with hypertension and T2D:

Treatment for diabetes is expensive. So, diabetes treatment is not given to people. See, when I came here the doctor prescribed me a pill that cost me 4,000 and some pesos. See people who are poor and do not work may be able to buy one box of medication but would not be able to buy two.

The above participant quote highlighted the dilemmas faced by people who have limited resources when managing their chronic conditions. This participant further described how T2D required money for "a proper diet," which made the need for balancing said proper diet and purchasing medication difficult when living in an economically vulnerable position. For participants that could not afford consistent access to specialists, a couple would go to government clinics (*policlínicas*) when needed for basic checkups.

The inconsistency of health care access, whether due to unclear referral system to establish care or lack of financial resources to see and build rapport with specialists, made it challenging for participants to manage their chronic condition. Structural barriers to care affected individual and interpersonal level factors impacting the co-management of multiple chronic conditions.

## Discussion

Adults experiencing food insecurity in the Dominican Republic who manage HIV and at least one chronic NCD encounter multi-level barriers and facilitators for appropriate co-management. Understanding these multi-level factors can improve our ability to develop appropriate programs and policies to improve the management of multiple conditions among adults.

In general, participants considered NCDs more difficult to manage than their HIV due to insufficient individual, interpersonal, and structural factors to facilitate care. Although all participants recently experienced moderate-to-severe food insecurity, they spoke of food insecurity when referencing managing chronic NCDs. Engaging in a clinician-recommended dietary regimen was challenging as some food items were not always available. In addition, participants spoke of general frustration with "not being able to eat anything anymore" because of their conditions, such as T2D and hypertension. This was a different narrative from HIV management where medication adherence was the most important self-management behavior discussed.

Individual factors influenced self-management across both conditions. For example, living a normal life was the most salient goal expressed by all participants. A qualitative study in the Dominican Republic found that the concept of "living a normal life" as related to HIV was indeed related to the notion of HIV becoming a "chronic condition" [37]. Nevertheless, in Barrington et al. [37], achieving a normal life was complicated by HIV-related stigma and discrimination and their implications for social relationships, and economic productivity, particularly for individuals living in resource-constrained areas. We found parallels in our study related to living a "normal life" with HIV. "Normal" in this case (for adults experiencing food insecurity) was attributed to their ability to consistently engage in treatment. In other words, the idea that they could live a long life with HIV equated to living a normal life. However, participants did speak about fears related to economic loss (e.g., not being able to work again) and rejection they faced from potential partners when they disclosed their HIV status, thus complicating their ability to develop social ties. In relation to chronic NCDs, striving to live a normal life was not brought up as consistently. This may have been due to the uncertainty around the management of chronic NCDs, which added a barrier to achieving the goal of living a normal life. Furthermore, there were parallels between how adults with HIV in our study and adults with T2D in the Dominican Republic in a recent qualitative study described the difficulties

associated with chronic conditions. Specifically, there was a perception that T2D was the worse condition compared to HIV because T2D required a multi-component self-management strategy that upended their entire lives, requiring not only medications, but also drastic changes in diet, and additional physical activity [30].

T2D was also considered a worse condition because of the stigmatizing stereotype attached to people with T2D, that they are quick to anger or get agitated (*pique*). Therefore, the need for people living with T2D to engage in emotional regulation was not only for their health but to counteract stereotypes. Concerns about the stereotypes associated with people with T2D were evident in other qualitative studies conducted in the Dominican Republic [30, 38]. When addressing interventions or programs that include people with T2D and HIV, ensuring that programs address the social stigmas associated with both conditions is necessary.

Interpersonal factors were focused on patient-provider relationships. All participants detailed a positive and productive relationship with their HIV providers. Although there was a lack of clear referrals for specialists to manage their chronic NCDs, additional barriers may emerge for PLHIV when seeking care for conditions outside of HIV clinics. A 2018 scoping review among PLHIV seeking to manage chronic NCDs found that fear of disclosure, internalized shame, negative past experiences with healthcare providers, or negative perceptions of healthcare providers, as well as HIV stigma from the part of providers all affected engagement in health care [22]. Therefore, public health researchers and practitioners must work to ensure that healthcare providers are trained on empathetic care and work to change any stigmatizing attitudes and perceptions they may have towards future patients who live with HIV, to ensure that patients are not confronted with stigmatizing environments.

Structural concerns focused on the healthcare system. Health insurance coverage was a prominent concern in participant narratives. Participants without health insurance did not seek care outside of the HIV clinic. Of note, all three Haitian-descendent participants did not have health insurance, which reflects their limited access to health insurance programs given the sociopolitical barriers this community faces such as the high cost of applying or renewing documentation (e.g., Dominican identification card) that is needed to apply for government insurance coverage [39, 40]. Participants who had health insurance noted similar structural barriers to seeking care, primarily the physical distance to seek specialists and the high out-of-pocket costs. Relatedly, participants spoke of the need to "save money" to seek care for non-HIV specialists or delay care due to the high costs of transportation, consults, and medication. The expectations of high out-of-pocket costs for health care were evidenced in the narratives of balancing financial needs and health care needs, particularly for chronic NCDs. This is reflective in a 2016 report in which the out-of-pocket costs at the point of service averaged at about 44.6% of the total cost of care in the Dominican Republic, which is at the higher end of the Americas region which averages 23.9% [41].

Relatedly, inequities in health services experiences were a fundamental barrier for self-management among participants. Participants spoke often about not receiving or not being aware of a referral to see a specialist for their condition or having their HIV provider offer follow-up for their non-HIV conditions. Contextually, these practice differences may be due to the capacities of the HIV program centers. There are some centers that have internal medicine doctors who can provide general follow-up to their HIV and other comorbidities; however, they will refer more severe cases to specialists at tertiary-level healthcare centers. This may explain why some participants noted that their HIV provider managed all aspects of their care. Other centers do not have more general medical practitioners available, therefore they only refer their patients to specialists at secondary-level hospitals. On the referral side, there is no system where HIV providers can communicate with specialists to follow-up with them regarding the status of any referred patients. The different methods used based on

center capacity and the barriers to following up on referrals, may have contributed to participant confusion on how to gain access to other forms of specialized care for their chronic NCDs. At a systems-level, improved standardization of the referral process and increased clarity of the healthcare process during patient visits may help HIV center patients better navigate managing their health. Furthermore, integrating the care of chronic NCDs within HIV care settings is a possible way of improving coordination between specialties, reducing barriers for patients seeking care, and improved holistic management for adults with multiple chronic conditions.

Medication affordability as a factor to engaging in chronic disease management also came up in a cross-sectional study by Castro and colleagues [8] who found that affordability of medications was a primary concern for Dominican adults living with hypertension or T2D. Cost concerns were not as large of an issue for HIV because the Dominican government provides all HIV medications and clinical care at no charge. However, similar programs are not available for chronic NCDs. Since 2001, laws were passed to promote universal health coverage in the public health sectors, which included preventive care and outpatient drug coverage. In relation to the health system, in 2006 the National Public Health Services (SENASA) constructed a public-private provider to improve financial accessibility to care [42]. For outpatient drug coverage, this program covers RD$8,000 pesos (USD$142) per year [43]; however, because of the high costs of medication, the coverage may only last two or three months. For general health services, in 2008, a plan was introduced to integrate chronic disease programs into the primary care model [8]. Shortly thereafter in 2009, the Ministry of Public Health developed the National Non-Communicable Chronic Disease Control and Prevention Program, aimed at the prevention and management of chronic NCDs and any associated risk factors [8]. In October 2015 an interinstitutional agreement was signed by CONAVIHSIDA, which is the National Council for HIV and AIDS, SENASA, and the Central Electoral Council to increase public health funds geared towards improved healthcare access (services, medications) for PLHIV [44]. The implications of this agreement allowed for PLHIV to request care for their HIV and other chronic conditions regardless of documentation status. The results of these policies are experienced in how participants described their ease of access to care for their HIV. These evolutions of health policies since the turn of the century indicate that health systems are responsive to the changing health priorities. Despite these improvements in health policy to address the increasing NCD burden, our findings show gaps in effectively implementing them at health service delivery level to improve care for people living with chronic NCDs. For instance, participants noted the ease of establishing and maintaining care for HIV; however, their experiences were the opposite for chronic NCDs highlighting an inequity in care.

## Strengths and limitations

The strengths of this study are that we explored the multilevel factors that affect co-managing multiple chronic conditions from the perspective of adult patients in the Dominican Republic, to our knowledge one of the first of its kind. Additionally, we focused our study on participants who recently experienced food insecurity, which centers the conversation on individuals who are managing their health amid economic vulnerabilities. Limitations of the study were that we selected and interviewed participants who were already linked to HIV care and were at the end of an intervention; therefore, the experiences of individuals who were currently not in HIV-related care were not reflected in these interviews. However, participants were able to reflect on their experiences getting linked to care, retrospectively, to provide a baseline comparison between their conditions.

## Conclusions

This study depicts the facilitators and barriers of managing multiple chronic conditions from the perspective of adult patients who recently experienced food insecurity in the Dominican Republic. As the number of adults living with HIV and other chronic conditions are increasing, we must contextualize these experiences to build future programs or policies that improve the health and well-being of individuals, particularly those who a navigating economic and social vulnerabilities. Individual self-management is necessary; however, it is not sufficient for the successful co-management of multiple chronic conditions. Interactions or relationships participants had with their health care team, improved referral systems, accessibility and availability of specialists, and financial support systems are important dimensions to assess in the development of health systems that account for the management of multiple chronic conditions.

## Supporting information

**S1 File. ProMeSA endline qualitative interview guide—English.**
(PDF)

**S2 File.**
(DOCX)

## Acknowledgments

The authors thank the participants for their generosity in providing their time as well as the health clinic staff that provided the space for the study. The authors would also like to thank Dr. Victor Terrero, Director of CONVIHSIDA, for ongoing support for our study and our efforts to understand and address food insecurity among people living with HIV. We thank colleagues from the Universidad Autónoma de Santo Domingo UASD and Ministry of Public Health Nutrition.

## Author Contributions

**Conceptualization:** Deshira D. Wallace, Kathryn P. Derose.

**Data curation:** Gipsy Jiménez Paulino, Flabia Tejada Castro, Stephanie Daniela Castro.

**Formal analysis:** Deshira D. Wallace, Gipsy Jiménez Paulino, Flabia Tejada Castro, Stephanie Daniela Castro.

**Funding acquisition:** Kathryn P. Derose.

**Investigation:** Amarilis Then-Paulino.

**Methodology:** Kartika Palar.

**Project administration:** Kathryn P. Derose.

**Supervision:** Deshira D. Wallace, Amarilis Then-Paulino, Kathryn P. Derose.

**Validation:** Kathryn P. Derose.

**Writing – original draft:** Deshira D. Wallace.

**Writing – review & editing:** Deshira D. Wallace, Amarilis Then-Paulino, Gipsy Jiménez Paulino, Flabia Tejada Castro, Stephanie Daniela Castro, Kartika Palar, Kathryn P. Derose.

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
