## [Decision Letter · Decision Letter 0]

28 Sep 2022

PONE-D-22-12189The co-management of HIV and chronic non-communicable diseases in the Dominican Republic: A qualitative studyPLOS ONE

Dear Dr. Wallace,

Thank you for submitting your manuscript to PLOS ONE. After careful consideration, we feel that it has merit but does not fully meet PLOS ONE’s publication criteria as it currently stands. Therefore, we invite you to submit a revised version of the manuscript that addresses the points raised during the review process. Please address all the comments from reviewers and provide point-to-point response.

We look forward to receiving your revised manuscript.

Kind regards,

Wenhui Mao, PhD

Academic Editor

PLOS ONE

Journal Requirements:

3. Please provide additional details regarding participant consent. In the ethics statement in the Methods and online submission information, please ensure that you have specified whether: 1) whether the ethics committee approved the verbal/oral consent procedure, 2) why written consent could not be obtained, and 3) how verbal/oral consent was recorded. If your study included minors, please state whether you obtained consent from parents or guardians in these cases. If the need for consent was waived by the ethics committee, please include this information.

Reviewers' comments:

Reviewer's Responses to Questions

**Comments to the Author**

1. Is the manuscript technically sound, and do the data support the conclusions?

Reviewer #1: Yes

Reviewer #2: Yes

2. Has the statistical analysis been performed appropriately and rigorously? 

Reviewer #1: N/A

Reviewer #2: N/A

3. Have the authors made all data underlying the findings in their manuscript fully available?

Reviewer #1: No

Reviewer #2: Yes

4. Is the manuscript presented in an intelligible fashion and written in standard English?

Reviewer #1: Yes

Reviewer #2: Yes

5. Review Comments to the Author

Reviewer #1: The co-management of HIV and chronic non-communicable diseases in the Dominican

Republic: A qualitative study

Review:

Manuscript presented an important topic and problem of high significance. The authors intended to explore PLWH lived experiences in co-managing multiple chronic conditions. There are significant and minor revisions suggested and described in each section.

Background

There is the mention of food insecurity in the background but not much is discussed about it. Is this a problem in Dominican Republic? Who is severely affected by food insecurity? Context can help guide the reason for including food insecurity.

Line 25-27 - “Of the individuals living with HIV in the Dominican Republic, 82% know their status and 51% of people living HIV, regardless of status, are on antiretroviral therapy (ART)”

This sentence is not clear. “Regardless of status” Does this mean that there are some without HIV and are accounted for in the 51%? Please assist the reader to understand the reason why it is important to include those without HIV and ART.

Line 31-38 described overall trends of NCDs with little mention of the current situation in Dominican Republic. For example, are the age groups of those with HIV changing to an older population? This can help motivate the significance of the research area.

Line 45-46: “At the individual level, those with multiple conditions require greater healthcare utilization and therefore incur greater healthcare costs than individuals with one or zero chronic conditions.”

Is this to be required or studies have found association between number of chronic conditions and higher utilization and cost?

Lines 39-69: Appreciated the background information but this section was hard to follow and also was too general. It would be helpful to narrow the focus to HIV and NCD. Discussing the conditions separately was very hard to tie back to the objective of the study. Please review this section.

Line 79 mentions the use of the socio ecological model to inform the study. In the background section, one expects to learn how the model has been used in other studies and then its choice is defended in the methods section. Was the model used to develop the interview guide? Why was this the model of choice?

Methods

Line 107- there is a mention of a “ProMeSA study” this is the first time it is mentioned and the last time. What was it? How was it related to the intervention?

Why did the researchers decide to interview participants in the intervention group? Since they were interested in barriers and facilitators for co-management of NCDs, what was specific about the intervention group?

Recommend use the COREQ (COnsolidated criteria for REporting Qualitative research) Checklist as a guide in what to be reported.

Results

Line 117: mentions of recruitment until saturation was reached. After how many participants was saturation reached?

Generally: although food insecurity is mentioned in the background, nothing seemed to have been explored in the topic.

Since the study was focused on co-management of HIV and chronic NCDs, it is important that the results mostly focus on this aspect. The sections beginning line 174 -239 seem to be out of the topic of discussion.

Minor: In table 2, authors should be consistent with abbreviations (T2D was used in some places and then one would see “type 2 diabetes” later)

Line 293 -294 “The lack of education was also tied to how participants experienced their condition as being “sensitive” and “unpredictable” from one day to the next.”

This sentence is not clear. Consider revising.

Line 327-337: this section was confusing. It is not clear what was said by participants.

Line 332-334 is also confusing. What information should the reader get from this sentence? “Of the five participants who were without health insurance …….three were the participants of Haitian descent”

Why was it important for the authors to add this sentence? Should the reader be introduced to the aspect earlier in the background?

Discussion

The discussion section provided information about that healthcare system that should have been provided in the background section. Consider revising this section to discuss the results and next steps.

Reviewer #2: Well done overall. The presentation and discussion on the key facilitators and barriers for individuals living with HIV and co-occurring NCDs was informative - especially Table 2.

Please re-read the manuscript as there are a few grammatical errors within the text.

6. PLOS authors have the option to publish the peer review history of their article (what does this mean?). If published, this will include your full peer review and any attached files.

Reviewer #1: No

Reviewer #2: No

---

## [Author Response · Author response to Decision Letter 0]

19 Nov 2022

Dear Editorial team,

We thank you for the opportunity to resubmit the manuscript: “The co-management of HIV and chronic non-communicable diseases in the Dominican Republic: A qualitative study” for exclusive consideration in PLOS One. 

Our response to reviewers is found below. 

Editorial Notes

• We have ensured that we followed the PLOS ONE style requirements. 

• The questionnaire on inclusivity in global research was completed and uploaded.

3. Please provide additional details regarding participant consent. In the ethics statement in the Methods and online submission information, please ensure that you have specified whether: 1) whether the ethics committee approved the verbal/oral consent procedure, 2) why written consent could not be obtained, and 3) how verbal/oral consent was recorded. If your study included minors, please state whether you obtained consent from parents or guardians in these cases. If the need for consent was waived by the ethics committee, please include this information.

• Consent information was updated. We specifically added text on why we conducted verbal consent (for literacy and accessibility purposes).

• We have added a statement on data availability of the following: “Because we did not state in the consent form that the data could be made available to outside researchers, our IRB has informed us that we cannot make the data available outside the study team.” 

• Therefore, we cannot provide a URL for public access to the data.

Based on the November 16th message from the PLOS One Editorial Team: 

Comment

1. We note your updated Data Availability statement as follows:

"Data underlying the results cannot be shared publicly to protect participant privacy and confidentiality. Because we did not state in the consent form that the data could be made available to outside researchers, our IRB has informed us that we cannot make the data available outside the study team."

In this instance it seems there may be acceptable restrictions in place that prevent the public sharing of your minimal data, however, in line with our goal of ensuring long-term data availability to all interested researchers, PLOS’ Data Policy states that authors cannot be the sole named individuals responsible for ensuring data access (http://journals.plos.org/plosone/s/data-availability#loc-acceptable-data-sharing-methods).

a) If there are ethical or legal restrictions on sharing a de-identified data set, please explain the restrictions in detail (e.g., data contain potentially identifying or sensitive patient information) and who has imposed them (e.g., a Research Ethics Committee or Institutional Review Board, etc.). Please also provide non-author contact information (phone/email/hyperlink) for a data access committee, ethics committee, or other institutional body to which data requests may be sent. If applicable, please also provide any necessary information which interested researchers would need when requesting access to data in order to obtain the minimal data set for your study.

Response

"The data upon which this manuscript is based are in-depth interview transcripts from a small sample of marginalized and stigmatized people with HIV and contain highly sensitive data that are identifiable via inference. Thus, sharing them would violate the promise of confidentiality made to participants during informed consent. Because we did not state in the consent form that the data could be made available to outside researchers, the RAND Human Subjects Protection Committee (HSPC) has informed us that we cannot make the data available outside the study team. Queries regarding data access can be addressed by the RAND HSPC by contacting Rebecca Collins, HSPC Chair (collins@rand.org)."

Comments to the Author

Reviewer #1:

 The co-management of HIV and chronic non-communicable diseases in the Dominican

Republic: A qualitative study

Review:

Manuscript presented an important topic and problem of high significance. The authors intended to explore PLWH lived experiences in co-managing multiple chronic conditions. There are significant and minor revisions suggested and described in each section.

Background

There is the mention of food insecurity in the background but not much is discussed about it. Is this a problem in Dominican Republic? Who is severely affected by food insecurity? Context can help guide the reason for including food insecurity.

• Thank you for this critique. We have added additional context and information about food security in lines 74-81. We included additional citations that focus on the role food insecurity has on the management of different conditions (HIV, NCDs).

Line 25-27 - “Of the individuals living with HIV in the Dominican Republic, 82% know their status and 51% of people living HIV, regardless of status, are on antiretroviral therapy (ART)”

This sentence is not clear. “Regardless of status” Does this mean that there are some without HIV and are accounted for in the 51%? Please assist the reader to understand the reason why it is important to include those without HIV and ART.

• The line in question was edited for clarity. 

Line 31-38 described overall trends of NCDs with little mention of the current situation in Dominican Republic. For example, are the age groups of those with HIV changing to an older population? This can help motivate the significance of the research area.

• Lines 34-37 are specifically referring to the overall trends in the Dominican Republic and lines 21-29 refers to the current situation in the Dominican Republic in regard to HIV. 

Line 45-46: “At the individual level, those with multiple conditions require greater healthcare utilization and therefore incur greater healthcare costs than individuals with one or zero chronic conditions.”

Is this to be required or studies have found association between number of chronic conditions and higher utilization and cost?

• We have added a citation to validate the argument. 

Lines 39-69: Appreciated the background information but this section was hard to follow and also was too general. It would be helpful to narrow the focus to HIV and NCD. Discussing the conditions separately was very hard to tie back to the objective of the study. Please review this section.

• Thank you for your comments. We reviewed this section and note that it is focused on the discussion of HIV and NCDs, specifically the need to consider the co-management of these conditions. Lines 17-37 do present the context of these conditions separately; however, the section indicates that the critique is focused on the interplay of these conditions. Therefore, we did not make any edits beyond those for clarity and readability. 

Line 79 mentions the use of the socio ecological model to inform the study. In the background section, one expects to learn how the model has been used in other studies and then its choice is defended in the methods section. Was the model used to develop the interview guide? Why was this the model of choice?

• The socioecological model is a framing for the data collection strategy and analysis. It’s been used extensively in public health, but to clarify for the reader on how we used it, we added a sentence to the Data Collection subsection that reads as follows: “Using a socioecological model we framed question to understand individual, interpersonal, organizational, and political barriers and facilitators to managing multiple chronic conditions.” Also, in lines 66-68 we reference the need to account for “individual, interpersonal, and structural facilitators and barriers to disease co-management…” which alludes to the socioecological model. 

Methods

Line 107- there is a mention of a “ProMeSA study” this is the first time it is mentioned and the last time. What was it? How was it related to the intervention?

• The ProMeSA study is the intervention study. We added language to clarify that the intervention study is named ProMeSA. Related information about the study was masked for review, however, we have cited the intervention for more information. 

• The endline interview are the data used for the present substudy, as explained in lines 109-117. 

Why did the researchers decide to interview participants in the intervention group? Since they were interested in barriers and facilitators for co-management 

of NCDs, what was specific about the intervention group?

• All participants were part of an intervention, thus there is not a differentiation between a non-intervention and intervention group. The intervention, ProMeSA, components was described in lines 95-101. We chose to do a qualitative evaluation of the intervention overall, but also asked about barriers and facilitators for co-managing multiple chronic conditions. Therefore, as the main study only targeted HIV-specific outcomes, we could ask about NCD self-management in the qualitative sub-study 

Recommend use the COREQ (COnsolidated criteria for REporting Qualitative research) Checklist as a guide in what to be reported.

• Thank you for the recommendation, we have filled out the COREQ checklist. 

Results

Line 117: mentions of recruitment until saturation was reached. After how many participants was saturation reached?

• At approximately 20 participants we did reach thematic saturation, therefore we stopped recruitment at 21 participants. 

Generally: although food insecurity is mentioned in the background, nothing seemed to have been explored in the topic.

• Thank you for this comment. The intervention study enrolled only participants who had moderate or severe food insecurity (per the Latin American Food Insecurity Scale). As this is the case, all narratives (i.e., data) are from this lens. We have added text in the results to this effect (lines 179-181). We also ensured to emphasize this throughout the manuscript to strengthen this foundation. 

Since the study was focused on co-management of HIV and chronic NCDs, it is important that the results mostly focus on this aspect. The sections beginning line 174 -239 seem to be out of the topic of discussion.

• Thank you for this comment. We understand that the focus is on the co-management of HIV and NCDs. We felt that it was important to provide some context for what it is like to live with HIV for adults who are food insecure in general, and equally what it is like to live with NCDs as an adult experiencing food insecurity. 

Minor: In table 2, authors should be consistent with abbreviations (T2D was used in some places and then one would see “type 2 diabetes” later)

• Thank you. We have corrected Table 2 for consistency. 

Line 293 -294 “The lack of education was also tied to how participants experienced their condition as being “sensitive” and “unpredictable” from one day to the next.”

This sentence is not clear. Consider revising.

• Thank you. We clarified what we meant by lack of education. The sentence now reads as follows: The lack of education on NCD self-management was also tied to how participants experienced their condition as being “sensitive” and “unpredictable” from one day to the next.

Line 327-337: this section was confusing. It is not clear what was said by participants.

• Thank you for your note. This first paragraph under structural level concerns focused on summarizing results from demographic variables (see Table 1). We also made a note throughout this paragraph that compared how participants spoke about their experiences with accessing the health care system. These experiences did not change much between those who were insured or not, particularly when managing NCDs. Therefore, we did not pull specific quotes. Rather we highlighted exemplar experiences for the remaining parts of the subsection. 

Line 332-334 is also confusing. What information should the reader get from this sentence? “Of the five participants who were without health insurance …….three were the participants of Haitian descent”

Why was it important for the authors to add this sentence? Should the reader be introduced to the aspect earlier in the background?

• At the beginning of the results section and Table 1, we have an ancestry break down in which three participants from our total pool of participants were of Haitian descent. We brought up this sentence of lack of health insurance in the results and further link to potential implications of this in second paragraph of the discussion section. In the discussion we note potential barriers for Haitians in the Dominican Republic to gain access to health insurance. We did not bring this up in the background because this was not an a priori assumption and it fit best to link the results to the discussion. 

Discussion

The discussion section provided information about that healthcare system that should have been provided in the background section. Consider revising this section to discuss the results and next steps.

• Thank you for this comment. The reason we have included the background information on the healthcare system in the discussion rather than in the background is because it was related to how participants described their experiences. This was not an a priori assumption, and we wanted to recognize this as linked to the data. 

Reviewer #2:

1. Well done overall. The presentation and discussion on the key facilitators and barriers for individuals living with HIV and co-occurring NCDs was informative - especially Table 2.

• Thank you for the comments. We appreciate that the manuscript was considered relevant and informative.

2. Please re-read the manuscript as there are a few grammatical errors within the text.

• We have re-read the manuscript and addressed grammatical errors.

---

## [Decision Letter · Decision Letter 1]

27 Mar 2023

PONE-D-22-12189R1The co-management of HIV and chronic non-communicable diseases in the Dominican Republic: A qualitative studyPLOS ONE

Dear Dr. Wallace,

Thank you for submitting your manuscript to PLOS ONE. After careful consideration, we feel that it has merit but does not fully meet PLOS ONE’s publication criteria as it currently stands. Therefore, we invite you to submit a revised version of the manuscript that addresses the points raised during the review process.

Please submit your revised manuscript by May 11 2023 11:59PM. If you will need more time than this to complete your revisions, please reply to this message or contact the journal office at plosone@plos.org. Please include the following items when submitting your revised manuscript:A rebuttal letter that responds to each point raised by the academic editor and reviewer(s). You should upload this letter as a separate file labeled 'Response to Reviewers'.A marked-up copy of your manuscript that highlights changes made to the original version. You should upload this as a separate file labeled 'Revised Manuscript with Track Changes'.An unmarked version of your revised paper without tracked changes. You should upload this as a separate file labeled 'Manuscript'.

We look forward to receiving your revised manuscript.

Kind regards,

Wenhui Mao, PhD

Academic Editor

PLOS ONE

Reviewers' comments:

Reviewer's Responses to Questions

**Comments to the Author**

1. If the authors have adequately addressed your comments raised in a previous round of review and you feel that this manuscript is now acceptable for publication, you may indicate that here to bypass the “Comments to the Author” section, enter your conflict of interest statement in the “Confidential to Editor” section, and submit your "Accept" recommendation.

Reviewer #3: (No Response)

Reviewer #4: (No Response)

2. Is the manuscript technically sound, and do the data support the conclusions?

Reviewer #3: Yes

Reviewer #4: (No Response)

3. Has the statistical analysis been performed appropriately and rigorously? 

Reviewer #3: N/A

Reviewer #4: N/A

4. Have the authors made all data underlying the findings in their manuscript fully available?

Reviewer #3: No

Reviewer #4: No

5. Is the manuscript presented in an intelligible fashion and written in standard English?

Reviewer #3: Yes

Reviewer #4: Yes

6. Review Comments to the Author

Reviewer #3: Thank you for a great manuscript providing key insights into a growing health challenge of high significance that is often understudied. With health systems grappling with an increasing double burden of communicable and non-communicable disease, co and multi morbid health conditions, the findings of the study provide key insights to informing health policy and planning within the context of competing resources. The paper is keenly needed but in its present form, i am sorry to say it leaves the reader confused. My comments and suggestions are attached in the manuscript. I hope this helps.

Reviewer #4: 1. Setting & participants: Were there any refusals? Any patients invited to participate who refused to consent?

Any bias: Was there a possibility of undue influence to participate if patients were recruited where they were getting care and the interventions?

2. Identification & recruitment: How exactly were the 21 patients:

- Identified e.g. through hospital datasets? Describe this step by step

-Invited to participate? E.g. via telephone calls? During their regular clinic visits? And so on

3. Verbal consent: The participants were engaged in-person. What does this mean ‘verbal consent to account for literacy and accessibility’?

How was the verbal consent carried out? Describe it step by step.

How was it captured by the research team?

4. 'Example questions included: how are you managing your HIV? How are you managing your chronic condition (T2D, high blood pressure)? Which of your health conditions worries you the most?' Provide the study tool as an appendage?

5. 'After each interview, interviewers wrote field notes about the interview setting, key themes of the interview, observations for the following interviews, and if any themes may have reached saturation.' Is it feasible to actually deduce data saturation at the end of each interview? How did the interviewers do it within short time spans? Add those details.

What was the lag between each interview?

6. Results. Very rich data reported. Thank you

7. Discussion points are appreciated, however, the flow can be improved. The ideas are there but the reader gets totally lost. Arrange the discussion to closely match the layout of the study findings

a.

-HIV management

-Chronic NCD management

b. Management of multiple conditions

-Individual

-Interpersonal

-Structural level

8. Nice conclusion. Edit to capture that these were '...Perspective of PATIENTS…not just 'adults experiencing food insecurity...'

9. Data availability. Can an interested researcher reach out to your corresponding author and/or their IRB if they have queries about your data and its analysis? Can you maintain possibility of future interactions with other researchers on data, associated tools, and data analysis?

10. Acknowledgment: 'The authors thank the participants for their generosity in providing their time as well as the health clinic staff and providers that participated.'

What was the role of health clinic staff and providers in this study? Were they involved in recruitment? Interviews? Any biases?

7. PLOS authors have the option to publish the peer review history of their article (what does this mean?). If published, this will include your full peer review and any attached files.

Reviewer #3: **Yes: **Dr. Gertrude Nsorma Nyaaba

Reviewer #4: **Yes: **Violet Naanyu

---

## [Author Response · Author response to Decision Letter 1]

27 Jun 2023

June 20, 2023

Dear Editorial Team and Reviewers,

We thank you for the opportunity to resubmit the manuscript: “The co-management of HIV and chronic non-communicable diseases in the Dominican Republic: A qualitative study” for exclusive consideration in PLOS One. 

Our response to reviewers is found below. 

Comments to the Author

Reviewer #3:

Thank you for a great manuscript providing key insights into a growing health challenge of high significance that is often understudied. With health systems grappling with an increasing double burden of communicable and non-communicable disease, co and multi morbid health conditions, the findings of the study provide key insights to informing health policy and planning within the context of competing resources. The paper is keenly needed but in its present form, i am sorry to say it leaves the reader confused. My comments and suggestions are attached in the manuscript. I hope this helps.

Reply:

• We thank the reviewer for their suggestions. We have addressed several comments and edited the manuscript to improve clarity, where needed. 

• Edits on Abstract:

o Food insecurity is an inclusion criterion as stated; therefore, the sentence was not edited. We kept the clause “we explored participant lived experiences co-managing…” because the focus of this manuscript is on the co-management piece, which is different from the inclusion criteria sentence. 

o Made small revisions to improve clarity and parallel structure to the abstract.

• Edits on Introduction:

o The “Introduction” is a header, not a title; therefore, it stayed as is to ensure we are in line with PLOS One’s template. 

o We understand that the reviewer believes that the introduction is long, however, we are providing the necessary context for the study. We preferred to keep the flow as is to reflect the flow of the results section, starting with HIV, moving into NCDs, and then gaps in information in co-management. 

o We addressed general edits to improve clarity. 

• Edits on Materials and Methods

o Thank you. We addressed grammatical errors and added some phrasing to improve clarity. 

o There were questions on some phrasing – RAND Corporation is the name of the company – RAND is not an acronym therefore we kept it as the official name of the organization. Semi-structured, in-depth interviews is the description of the methods. For example, in-depth interviews means one-on-one interviews. 

o Prior reviewers requested example questions and those were provided. Interview guide is in Spanish (not translated); however, we have attached the English version of the interview guide. 

o Data collection processes were described according to COREQ guidelines. 

o We placed the number of interviews (21) in the results. 

• Edits on Results

o None.

• Edits on Discussion

o Made line edits to improve clarity throughout the document. Rearranged the discussion to focus on themes in a similar order to the results section to improve clarity.

Reviewer #4:

Setting & participants: Were there any refusals? Any patients invited to participate who refused to consent?

Any bias: Was there a possibility of undue influence to participate if patients were recruited where they were getting care and the interventions?.

• Thank you for the comments. Yes, participants were able to refuse to participate in the study. Through the consent process, patients also got assurances that refusal to participate would not affect their care. Specific language in the consent is: “Your participation in this interview is completely voluntary. You do not have to participate if you do not want to and you can stop the interview at any time or withdraw your consent to be in the study for any reason without penalty or repercussions.”

2. Identification & recruitment: How exactly were the 21 patients:

- Identified e.g. through hospital datasets? Describe this step by step

-Invited to participate? E.g. via telephone calls? During their regular clinic visits? And so on

• Thank you for your questions. Participants were identified through the sampling frame from the intervention and based on clinic records. We provide details of this under settings and participants: “For the qualitative endline sub-study, we used in-person purposive sampling to select intervention participants who also had a chronic NCD (i.e., hypertension, diabetes, prediabetes, high cholesterol, obesity) according to previously collected data. We purposively recruited approximately equal numbers of men and women.”

3. Verbal consent: The participants were engaged in-person. What does this mean ‘verbal consent to account for literacy and accessibility’?

How was the verbal consent carried out? Describe it step by step.

How was it captured by the research team?

• Thank you. Verbal consent rather than written consent was done to account for variability in literacy (i.e., not everyone can read). To be explicit about that, we are leaving the phrasing as is. 

• Verbal consent was carried out in person, the consent form read aloud, and at the end the participant could verbally indicate their consent to participate. 

4. 'Example questions included: how are you managing your HIV? How are you managing your chronic condition (T2D, high blood pressure)? Which of your health conditions worries you the most?' Provide the study tool as an appendage?

• We do include sample questions, and have included an English version of the guide in the appendix. 

5. 'After each interview, interviewers wrote field notes about the interview setting, key themes of the interview, observations for the following interviews, and if any themes may have reached saturation.' Is it feasible to actually deduce data saturation at the end of each interview? How did the interviewers do it within short time spans? Add those details.

• Field notes are always done immediately after each interview. We added the word “immediately” to clarify time lag. 

• We simplified the language and removed themes reaching saturation to better reflect the ana

• We are unsure what the question is asking about how interviewer do within short time spans. Does this refer to the field notes or to how many interviews are done? The number of interviews done were based on the day and clinic and we indicate that the 21 interviews were conducted over the course of 4 months (October 2019-February 2020). 

6. Results. Very rich data reported. Thank you

• Thank you, we appreciate that. 

7. Discussion points are appreciated, however, the flow can be improved. The ideas are there but the reader gets totally lost. Arrange the discussion to closely match the layout of the study findings

a.

-HIV management

-Chronic NCD management

b. Management of multiple conditions

-Individual

-Interpersonal

-Structural level

• Thank you for your comments. The discussion section was revised as needed. 

8. Nice conclusion. Edit to capture that these were '...Perspective of PATIENTS…not just 'adults experiencing food insecurity...'

• Edited in the strengths and conclusions section. 

9. Data availability. Can an interested researcher reach out to your corresponding author and/or their IRB if they have queries about your data and its analysis? Can you maintain possibility of future interactions with other researchers on data, associated tools, and data analysis?

• Yes, inquiries can be addressed on a rolling basis. The research team is happy to talk through the analysis and discuss data structure. However, given that IRB and ethics approvals were needed across countries and different institutions, we cannot place the data in a repository to be widely shared, as it violates ethics agreements. 

10. Acknowledgment: 'The authors thank the participants for their generosity in providing their time as well as the health clinic staff and providers that participated.'

What was the role of health clinic staff and providers in this study? Were they involved in recruitment? Interviews? Any biases?

• No, health clinic staff were not involved in the recruitment. We clarified this in the acknowledgements. The research staff were independent of the clinic. We thank the health staff for allowing the team space to conduct the research.

---

## [Editor Report · Decision Letter 2]

2 Jul 2023

The co-management of HIV and chronic non-communicable diseases in the Dominican Republic: A qualitative study

PONE-D-22-12189R2

Dear Dr. Wallace,

We’re pleased to inform you that your manuscript has been judged scientifically suitable for publication and will be formally accepted for publication once it meets all outstanding technical requirements.

Kind regards,

Wenhui Mao, PhD

Academic Editor

PLOS ONE

---

## [Editor Report · Acceptance letter]

5 Jul 2023

PONE-D-22-12189R2 

The co-management of HIV and chronic non-communicable diseases in the Dominican Republic: A qualitative study 

Dear Dr. Wallace:

I'm pleased to inform you that your manuscript has been deemed suitable for publication in PLOS ONE. Congratulations! Your manuscript is now with our production department. 

Kind regards, 

on behalf of

Dr. Wenhui Mao 

Academic Editor

PLOS ONE